# Unveiling the chain mediation model of sleep quality and fatigue in linking subjective exercise experience to depression in adolescent

Xiaoyu Li[1], Qian Yang[1☢], Ziyang Qi[2☢], Changhao Ma[1☢], Shenghua Qi[1]*, Wenliang Ji[1]*

1 School of Physical Education, University of Jinan, Jinan, Shan Dong, China, 2 School of Education and Psychology, University of Jinan, Jinan, Shan Dong, China

☢ These authors contributed equally to this work.
* spe_qish@ujn.edu.cn (SQ); spe_jiwl@ujn.edu.cn (WJ)

## Abstract

While the positive correlation between physical exercise and depression has been widely replicated in cross-sectional research, the prognostic value of subjective exercise experience for depression continues to be disputed, and the mediating processes are not well understood. This study aims to elucidate the predictive role of subjective exercise experience in depression and examine the chain mediating effects of sleep quality and fatigue. We administered validated self-report questionnaires to assess these constructs in a sample of 538 Chinese high school students. The results demonstrated that adolescents' depressive symptoms are influenced by subjective exercise experience, and this effect mediated through two independent pathways: sleep quality and fatigue. Moreover, subjective exercise experience can predict adolescent depression via either the sleep quality pathway or the fatigue pathway. The identification of mediating dual pathways provides empirical support for developing targeted exercise interventions that specifically address sleep disturbances and fatigue management in depressed adolescents.

## 1 Introduction

Depression has emerged as one of the most pervasive and clinically significant mental health disorders affecting adolescent populations worldwide [1]. Major depressive disorder, characterized by its chronic and relapsing nature, demonstrates particularly concerning epidemiological patterns in youth populations [2,3]. Current evidence suggests that nearly 70% of affected individuals endure recurrent episodes throughout their lifespan, while 40–60% experience relapses within just two years post-diagnosis [4]. Clinically, adolescent major depressive disorder presents with three concerning characteristics: progressively earlier age of onset, subtle symptomatic manifestations that often evade detection, and remarkably persistent clinical courses [5–7]. The adolescent developmental period represents a vulnerable neurobiological and

**Data availability statement:** All data are in the manuscript and/or supporting information files.

**Funding:** This work was supported by the National Social Science Fund of China (NSSFC) grant number [21BTY004]. The funder contributed to the initial conceptualization of the study by providing research ideas and decisions to publish. The funder had no role in data collection and analysis, interpretation of the results or preparation of the manuscript.

psychosocial transition phase marked by rapid physiological maturation, psychological reorganization, and social role transformation [8]. This perfect storm of biological vulnerability and psychosocial instability creates heightened susceptibility to various psychopathologies such as depression, anxiety disorders, anhedonia, low self-worth and sleep disturbances [9]. The resultant psychological fragility underscores the critical need for mechanistic research into modifiable risk factors and the development of evidence-based non-pharmacological interventions [10]. Addressing this pressing public health challenge requires urgent investigation into innovative prevention and treatment strategies capable of breaking the cycle of chronicity that characterizes adolescent-onset depression.

Numerous studies have demonstrated that subjective exercise experience is positively correlated with alleviating depressive symptoms [11]. Among adolescents, positive perceptions of exercise such as enjoyment, satisfaction, and a sense of accomplishment have been associated with lower levels of depression [12]. Randomized controlled trials indicate that engaging in moderate-intensity physical activity that is experienced positively can lead to significant reductions in depressive mood and emotional distress. Cross-sectional research conducted in various countries suggests that both aerobic and relaxation-based exercise, when perceived positively, are effective in alleviating depressive symptoms, particularly combined with healthy lifestyle behaviors such as sufficient sleep and reduced sedentary time [13]. Overall, fostering positive subjective exercise experiences appears to be a promising strategy for mitigating depression among adolescents across diverse populations.

Despite the extensive body of research investigating the impact of physical activity on mental health, the underlying mechanisms that connect subjective exercise experience with depressive symptoms remain largely underexplored and poorly understood. Meta-analytic evidence indicates that lower levels of physical activity or negative exercise experiences are associated with a higher risk of depression, which may also contribute to the onset of fatigue and other adverse physiological outcomes [14]. Moreover, lifestyle factors, such as excessive screen use or irregular daily routines, can exacerbate sleep disturbances and emotional dysregulation, further increasing vulnerability to depression [15]. In contemporary adolescence, pervasive exposure to electronic devices and academic pressures has heightened concerns regarding sleep quality, fatigue, and emotional well-being [16]. It is suggested that enhancing adolescents' self-regulation and promoting positive subjective exercise experiences may buffer against depressive symptoms. Regular engagement in enjoyable and satisfying physical activity has been shown to improve mood, reduce fatigue, and support adaptive coping strategies [17]. This raises a critical question: whether the reduction of depressive symptoms through subjective exercise experience is achieved by improving sleep quality and alleviating fatigue. Empirical investigation of this sequential mechanism is warranted to clarify the behavioral-physiological-emotional pathways involved. Therefore, we conducted a cross-sectional study among Chinese adolescents to explore the mechanisms linking subjective exercise experience and depressive symptoms. The findings of this study are expected to advance the current understanding of how positive exercise

experiences relate to mental health, and to provide practical guidance for interventions aimed at enhancing adolescent mental health through optimizing exercise experiences, sleep quality, and fatigue management.

## 1.1 Correlations between subjective exercise experience and depression

Subjective exercise experience refers to an individual's internal perception and affective response during physical activity, serving as a key predictor of sustained exercise behavior. It encompasses the subjective sensations and emotional reactions elicited by participation in exercise, including dimensions such as positive effect (e.g., enjoyment and satisfaction), psychological distress, and perceived fatigue. Physical exercise has gained recognition as a dual-purpose intervention, offering both physiological benefits and psychological regulation, thereby emerging as a crucial lifestyle approach for mitigating adolescent depression [18,19]. A substantial body of evidence demonstrates that regular physical activity not only improves cardiovascular fitness and metabolic health but also enhances emotional regulation, subjective well-being, and psychological resilience in youth populations [20,21]. However, the current research remains predominantly focused on quantifiable behavioral metrics including exercise frequency, duration and intensity, while largely neglecting the individuals' subjective experiences and emotional responses during physical activity engagement [22,23]. Empirical studies suggest that positive exercise experiences can enhance an individual's intrinsic motivation and foster greater engagement in physical activity, subsequently improving affective states and stress coping mechanisms that potentially serve as protective factors against depressive symptomatology [11,24,25]. Nevertheless, significant knowledge gaps persist regarding: (1) the longitudinal predictive validity of subjective exercise experiences for depression risk, and (2) the precise mechanistic pathways underlying this relationship. Accordingly, we propose that subjective exercise experience can significantly predict sleep quality (Hypothesis 1).

## 1.2 The mediating role of sleep quality

Good sleep quality is essential for maintaining normal physiological and psychological functioning. Prolonged sleep disturbances, however, can contribute to significant physical and mental health impairments. As a critical physiological pillar of mental well-being, sleep quality plays a fundamental role in emotional regulation and psychological recovery. Sleep disturbances often serve as an early indicator of emotional disorders, particularly depression, and pose a significant risk to both physical and mental health. Epidemiological studies reveal that individuals with poor sleep quality are markedly more likely to exhibit depressive symptoms compared to those with normal sleep patterns-with a 4.5-fold increased risk of developing depression [26]. Existing research demonstrates that positive exercise experiences contribute to improved sleep quality, which in turn mitigates negative emotions [27]. Therefore, sleep quality may serve as a mediating variable in the relationship between subjective exercise experiences and depression (Hypothesis 2).

## 1.3 The mediating role of fatigue

Fatigue represents a multidimensional state of profound physical and mental exhaustion, characterized by persistent lethargy, depleted energy reserves, and generalized weakness [28]. As a nonspecific yet complex subjective symptom, it exerts significant psychological burden on individuals, potentially culminating in affective disturbances such as depression [29,30]. According to the stress susceptibility model, potentially serving as both a precursor and perpetuating factor in mood disorders, fatigue exacerbates an individual's vulnerability to disease pathogenesis and progression while elevating risks of physiological and psychological dysregulation [31]. Furthermore, Fatigue, whether precipitated by sleep disturbances or compounded by chronic emotional stress and sustained energy expenditure, substantially impairs psychological resilience and diminishes emotional regulation capacity [32]. Many studies demonstrate that positive emotions in subjective exercise experience can effectively mitigate both physical fatigue and mental exhaustion [11]. These beneficial emotional experiences contribute to sustained energy maintenance, enhanced psychological homeostasis and decreased

depressive symptomatology. Hence, we propose that fatigue mediates the relationship between subjective exercise experience and depression (Hypothesis 3).

### 1.4 Chain mediation of sleep quality and fatigue

Sleep quality and fatigue represent interrelated psycho physiological processes that exhibit bidirectional influence and shared mechanistic pathways. Sleep quality and fatigue as critical parallel and sequential mediators linking subjective exercise experience to depressive symptoms in adolescents. Diminished sleep quality consistently correlates with increased fatigue severity, and sustained fatigue subsequently impairs sleep architecture, generating a mutually reinforcing pathological cycle [33,34]. Consequently, sleep quality may act as a mediating variable that links subjective exercise experience to depressive symptoms. Additionally, it may indirectly affect emotional health by inducing fatigue, thereby establishing a chain mediation pathway from sleep quality to fatigue (Hypothesis 4).

While prior research has examined the link between subjective exercise experience and depression, the underlying mechanisms- particularly the mediating roles of sleep quality and fatigue remain insufficiently explored [35]. To address this issue, this study aims to investigate the relationship between subjective exercise experience, sleep quality, fatigue, and adolescent depression. Specifically, we will assess whether sleep quality and fatigue serve as mediators in the association between subjective exercise experience and depressive symptoms in adolescents. By analyzing these potential mediating pathways, this study seeks to elucidate the psychological mechanisms underlying non-pharmacological interventions. Ultimately, this finding may contribute to a deeper theoretical understanding and inform more effective strategies for the prevention and intervention of adolescent depression.

## 2 Methods

### 2.1 Participants and procedure

Participants were recruited from June 3 to June 30, 2025, from Shandong Province, China. The studies involving human participants were reviewed and approved by the Ethics Committee of the School of Physical Education, University of Jinan (〔2025〕) on June 3, 2025. The study was conducted in accordance with the principles of the Declaration of Helsinki, as revised in 2013. This study adopted a stratified cluster random sampling method, with adolescent students from three middle schools (including key high schools and regular high schools) in Binzhou City, Shandong Province, as the sampling frame. First, stratification was conducted by school, followed by random selection of intact classes from each grade as the minimum sampling units. The experimental manipulations were approved the guidelines of Science and Technology Office of University of Jinan (2025000096). Informed consent was obtained from all individuals approved of incitants included in the study. A total of 570 paper questionnaires were distributed, with 536 valid responses collected after excluding invalid cases (including incomplete responses exceeding 20%, multiple selections for the same option, or inconsistent logic). The effective response rate reached 94.04%. Among the participants, 274 (51.1%) identified as male and 262 (48.9%) as female. All respondents completed the questionnaire voluntarily after providing informed consent. The study adhered to ethical guidelines for psychological research, with written informed consent obtained from both participating students and their legal guardians.

### 2.2 Research design and variable setting

This study aims to examine the potential mediating effects of sleep quality and fatigue perception on the relationship between subjective exercise experience and depression among adolescents, thereby establishing and validating a chain mediation model. The conceptual framework included: an independent variable (subjective exercise experience), two sequential mediators (sleep quality and fatigue), and a dependent variable (depression). All constructions were assessed using psychometrically validated scales, with higher composite scores reflecting more pronounced manifestations of each respective variable.

## 2.3 Research tools

Using the structural equation model (SEM) and AtongjiMOS 27.0, the researchers employed Anderson's two-step methodology [36]. First, the validity of the measurement model is evaluated through confirmatory factor analysis (CFA). Second, SEM was performed to assess model fit and examine path coefficients. During the CFA stage, indicators with standardized factor loadings below 0.5 were removed to ensure robust measurement of the latent constructs and improve model fit. Additionally, residuals of indicators are not allowed to be interrelated. Problematic indicators should be excluded to improve the fitting of the measurement model and create strong latent variables. According to the fitting criteria, if $x^2/df$ is less than 5, RMSEA is less than 0.08, and NFI, CFI, and GFI are greater than 0.90, it is determined that the model fits well.

**2.3.1 Subjective exercise experience scale (SEES).** The SEES was used to assess subjective exercise experience at (McAuley et al.,1994) [37]. The scale consists of 12 items and includes 3 dimensions: positive well-being, psychological distress and fatigue. Each dimension consists of 4 items and uses a 7-point Likert score (1 = very inconsistent, 7 = very consistent). Among them, the dimension of positive well-being is scored positively, while psychological distress and fatigue are scored negatively. Higher total scores indicate more positive post-exercise subjective experiences. Following CFA validation standards, all items were retained for structural equation modeling analysis. The model showed $X^2 = 242.504$, df = 51, $x^2/df = 4.755$, NFI = 0.939, GF1 = 0.930, CF1 = 0.951, RMSEA = 0.084. The Cronbach's α value for this scale reached 0.703 in our study.

**2.3.2 Depression scale (PHQ-9).** The PHQ-9 is a widely used depression screening tool with excellent validity and reliability [38]. This 9-item scale evaluates the frequency of depressive symptoms in participants over the past two weeks, using a 4-point rating scale (0 = none, 3 = almost daily). Total scores range from 0 to 27, where higher scores indicate more severe depression. In accordance with CFA validation standards, all items were retained for final structural equation modeling analysis. The model showed significant results: $X^2 = 109.337$ (df = 26), $X^2/df = 4.205$, NFI = 0.957, GF1 = 0.956, CF1 = 0.967, RMSEA = 0.077. The Cronbach's α value for this scale in our study was 0.908.

**2.3.3 Pittsburgh sleep quality index (PSQI).** The PSQI was employed to evaluate individual subjective sleep quality, comprising 7 dimensions and 16 items covering aspects such as time to fall asleep, sleep efficiency, and daytime functioning. Higher scores indicate poorer sleep quality [39]. Exploratory factor analysis revealed the KMO value was 0.926, and the Bartlett sphericity test was significant (P < 0.001), which met the conditions for factor analysis. Through principal component analysis and Varimax rotation, factors with eigenvalues greater than 1 were extracted, ultimately yielding two factors with eigenvalues of 6.634 and 1.194. The study utilized the Chinese version, with an internal consistency Cronbach's α of 0.901.

**2.3.4 Fatigue rating scale (FAI).** The FAI (Joseph E. Schwar et al., 1993) comprises 29 items divided into four factors: perceived fatigue severity, environmental sensitivity, fatigue outcomes, and rest response [40]. The scale employs a 7-point Likert scale where higher scores indicate more severe and context-specific fatigue experiences. Exploratory factor analysis revealed a KMO value of 0.978 and significant Bartlett's test of sphericity (P < 0.001), meeting criterion-fitting requirements. Principal component analysis combined with Varimax rotation identified two factors with eigenvalues greater than 1, yielding eigenvalues of 19.246 and 1.049 that collectively explained 69.982% of total variance. The Cronbach's α coefficient reached 0.982, demonstrating exceptional measurement stability in this study.

## 2.4 Statistical analysis

Spss27.0 statistics was used for data processing. The statistical description mainly used the mean plus or minus standard deviation to represent the distribution of measurement values. Harman was used to test the common method deviation, Pearson was used for correlation analysis, and AMOS27.0 was used to test the chain mediating effect. Chain mediation effect analysis is designed to reveal indirect mechanisms through which independent variables influence dependent variables via multiple interconnected mediating variables. This refers to the sequential transmission of independent variable

(X) through two or more mediating variables (M1, M2...) to produce an indirect impact on dependent variable (Y). In the chain model, subjective exercise experience is taken as the independent variable, and adolescent depression is regarded as the dependent variable. To enhance the stability of mediation effect testing, bootstrap (n = 5000) method was used for standard error correction, and confidence intervals with bias adjustments were employed to determine significance. Specifically, when the 95% confidence interval (95% CI) of the mediation effect contains zero, it is considered insignificant; if it excludes zero, it is deemed significant [41].

## 3 Results

### 3.1 Common method bias test

Using Harman single-factor test to analysis for common method bias, exploratory factor analysis was conducted on all items across the scales [42]. The results showed that when unrotated, eight factors with eigenvalues greater than 1 were identified. The first factor explained 44.16% of the variance, which is below Hair' recommended threshold of 50%, indicating that the common method bias in this study is not significant [43].

### 3.2 Descriptive statistics and normality test

Before the formal structural modeling analysis, this study first conducted descriptive statistics and normality test on four core variables: subjective exercise experience, sleep quality, fatigue and depression. As shown in Table 1, the skewness of each variable is within the range of ±2, and the kurtosis is not more than ±7, indicating that each variable is close to the normal distribution and meets the prerequisite conditions for structural equation model analysis.

### 3.3 Correlation analysis between variables

To verify the correlation direction and intensity between variables and provide a basis for model construction, Pearson correlation analysis was conducted in this study. Table 1 presents the correlation analysis among subjective exercise experience, sleep quality, fatigue, and depression. Subjective exercise experience showed significant positive correlations with depression (r = 0.335, p < 0.001), fatigue (r = 0.340, p < 0.001), and sleep quality (r = 0.288, p < 0.001). Depression is significantly positively correlated with fatigue (r = 0.728, p < 0.001) and sleep quality (r = 0.718, p < 0.001). Fatigue is significantly positive correlation with sleep quality (r = 0.781, p < 0.001). There was a significant positive correlation between subjective exercise experience, sleep quality, fatigue and depression. The correlation coefficients between sleep quality and fatigue, as well as fatigue and depression, both exceed 0.7, indicating that there was a strong variable relationship and had the theoretical basis to construct a chain mediation model.

### 3.4 Structural equation model analysis

We conducted a mediation effect analysis using Model 6 from SPSS PROCESS plugin. The chain mediation model incorporated subjective exercise experience as the independent variable, depression as the dependent variable, and sleep

**Table 1. Descriptive statistics of variables (N = 536).**

| Variables | M | SD | Skewness | Kurtosis |
|---|---|---|---|---|
| Subjective exercise experience | 41.709 | 8.99871 | 0.652 | 2.926 |
| Depression | 13.0392 | 4.93664 | 1.556 | 2.834 |
| Fatigue | 58.5373 | 26.94755 | 0.774 | −0.04 |
| Sleep quality | 25.653 | 9.01478 | 1.098 | 0.905 |

quality and fatigue as mediating variables. All variables were standardized prior to analysis. Results showed that all path coefficients in the model reached statistical significance (Table 2).

Based on the theoretical model and previous analysis results, this study constructed a structural equation model with subjective exercise experience as the predicted variable, sleep quality and fatigue as the mediating variables, and depression as the dependent variable. The structural model was established and validated using AMOS 27.0. As shown in Table 3, the initial model fitting index is a chi-square free ratio ($\chi^2$/df) of 3.513 and a root mean square error of the incremental residual sum (RMESA) of 0.069. The evaluation criteria for model fit: a $\chi^2$/df value below 5 and a RMESA less than 0.08.

### 3.5 Relationship between subjective exercise experience and depression: a chain mediating test

As shown in Fig 1, the subjective exercise experience showed a significant positive correlation with sleep quality ($\beta = 0.2882$, $p < 0.001$). There was a significant positive correlation between sleep quality and fatigue ($\beta = 0.7450$, $p < 0.001$). Subjective exercise experience ($\beta = 0.125$, $p < 0.001$) and fatigue ($\beta = 0.4017$, $p < 0.001$) were significantly positively correlated with depression, while sleep quality was significantly positively correlated with depression ($\beta = 0.3790$,

**Table 2. Regression analysis of variable relationships in chain mediation model.**

| Variables | Sleep quality | | Fatigue | | Depression | |
|---|---|---|---|---|---|---|
| | β | t | β | t | β | t |
| Subjective exercise experience | 0.2882 | 0.0415*** | 0.125 | 4.5077*** | 0.0891 | 3.0346** |
| Sleep quality | | | 0.745 | 26.8709*** | 0.379 | 8.5650*** |
| Fatigue | | | | | 0.4017 | 8.9166*** |
| R² | 0.0831 | | 0.6243 | | 0.5944 | |
| F | 48.3713*** | | 442.874*** | | 259.9306*** | |

**p<0.01, ***p<0.001.

**Table 3. The goodness of fit index of the model.**

| X² | df | CMIN/DE | RMSEA |
|---|---|---|---|
| 7282.404 | 2073 | 3.513 | 0.069 |

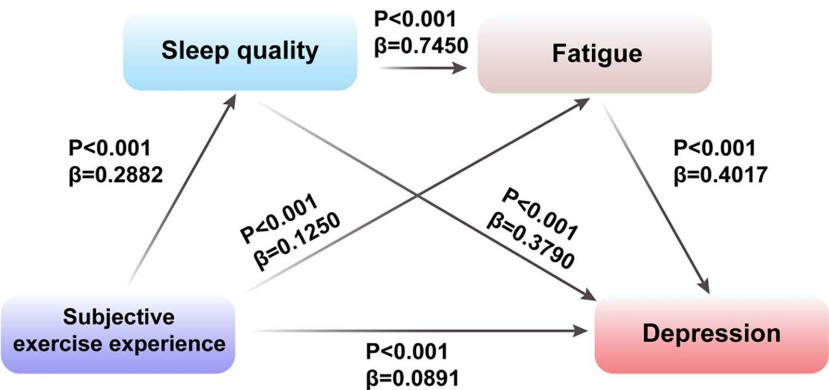

**Fig 1. The mediating role of sleep quality and fatigue in the relationship between subjective exercise experience.**

p<0.001). Using Bootstrap method for bias correction, this study calculated the mediation effect and its 95% confidence interval through 5000 repeated sampling. If the confidence interval does not include the zero value, it indicates statistical significance (Table 2).

### 3.6 The relationship between subjective exercise experience and depression: chain mediation effect test

The results (Table 4) showed that: the bootstrap 95% confidence interval for the total indirect effect (0.1348) of perceived sleep quality and fatigue did not contain a value of 0, indicating the two mediators had a significant mediating effect between sleep quality and depression. The total indirect effect is composed of three indirect effects: (1) indirect effects caused by subjective exercise experience duration-sleep quality-depression (0.0599); (2) indirect effects of subjective exercise experience duration-fatigue-depression (0.0275); (3) indirect effects of subjective exercise experience duration-sleep quality-fatigue-depression (0.0473).

These results demonstrated that the impact of adolescents' subjective exercise experience on depressive tendencies manifests through both direct and mediating effects, with the direct effect accounting for 73.38% and the mediating effect contributing 25.75% of the total effect. As shown in Table 4, the 95% confidence intervals for the three pathways are [0.0328, 0.0941], [0.0083, 0.0458], and [0.0269, 0.0714] respectively, indicating significant mediating effects across all pathways. These findings suggest that mediating effects operate through three distinct pathways: subjective exercise experience-sleep quality-depression, subjective exercise experience-fatigue-depression, and subjective exercise experience-sleep quality-fatigue-depression.

## 4 Discussion

### 4.1 Direct pathways from subjective exercise experience to depression

Our findings identify subjective exercise experience as an important factor for influencing depression that underscore the preventive value of interventions aimed at improving subjective motor experiences to mitigate depression risk among adolescents. Adolescent subjective exercise experience encompasses three dimensions: positive well-being, psychological distress, and fatigue [44]. Physical exercise is positively correlated with the subjective well-being of college students. When positive subjective experiences such as pleasure, relaxation and satisfaction occur during exercise, the brain secretes neurotransmitters [45]. For example, endorphins act as natural painkillers, enhancing euphoria and excitement to alleviate tension and anxiety while reducing negative emotions [46]. Dopamine boosts pleasure and well-being, helping individuals overcome depressive moods. Promoting the acquisition of positive well-being after exercise and inhibiting psychological distress and fatigue may positively contribute to reducing adolescent depression tendencies [47]. This study validated Hypothesis 1. However, positive well-being is influenced by exercise behavior [48]. Moderate-intensity exercise reduces the use of antidepressant medication, while higher intensity and longer duration of exercise correlate with lower

**Table 4. Bootstrap test results of mediation effect.**

| Items | Effect | BootSE | BootLLCI | BootULCI | The Proportion of Effect Size |
|---|---|---|---|---|---|
| Direct effect | 0.0489 | 0.0161 | 0.0172 | 0.0806 | 26.62% |
| Ind1 | 0.0599 | 0.0159 | 0.0328 | 0.0941 | 32.61% |
| Ind2 | 0.0275 | 0.0083 | 0.0134 | 0.0458 | 14.97% |
| Ind3 | 0.0473 | 0.0113 | 0.0269 | 0.0714 | 25.75% |
| Total indirect effects | 0.1348 | 0.0241 | 0.0898 | 0.1838 | 73.38% |

Indirect effect 1: subjective exercise experience-sleep quality-depression; Indirect effect 2: subjective exercise experience-fatigue-depression; Indirect effect 3: subjective exercise experience-sleep quality-fatigue-depression.

levels of positive well-being. Therefore, maintaining moderate exercise intensity and duration better facilitates the suppression of depression.

## 4.2 Mediating mechanism of sleep quality

Sleep quality serves as a mediating mediator in the relationship between subjective exercise experience and depression [49], and thus Hypothesis 2 was validated. Sleep deprivation disrupts circadian regulation in the hypothalamic suprachiasmatic nucleus, establishing a neurobiological pathway for depression development. This pathological process manifests through multiple mechanisms, including upregulated inflammatory markers and dysregulated neurotransmitter systems. Individuals with insufficient physical activity are more prone to sleep disorders, which may subsequently precipitate depressive symptomatology [50]. As sleep quality deteriorates, chronic fatigue emerges, further reducing motivation for physical activity. This bidirectional deterioration creates a self-perpetuating cycle that amplifies depression risk [51]. Functioning as a core determinant of mental health, sleep quality not only shows significant positive correlation with depression severity and progression but may also directly or indirectly elevate depressive risks through fatigue pathways [52]. Beneficial physical activity enhances adolescent sleep quality, while adequate nocturnal recovery facilitates the normalization of emotional regulation. From a psychological perspective, exercise-induced positive affect mitigates sleep difficulties and disrupts the anxiety-sleep disorder cycle. High-quality sleep confers multiple advantages: maintaining energy homeostasis, modulating stress responses, stabilizing mood, optimizing cognitive performance, and promoting overall psychological well-being. For adolescents, academic stressors and irregular sleep patterns frequently result in sleep deprivation and phase delay. Strategic enhancement of exercise experience-promoting emotional decompression, physiological fatigue, and circadian realignment represents an effective intervention to improve sleep quality that may substantially attenuate the detrimental effects of sleep disturbances on depression.

## 4.3 The mediation and chain mediation path of fatigue

Our study revealed that fatigue served as a critical mediating factor in the relationship between subjective exercise experience and depression, thus supporting Hypothesis 3. Specifically, fatigue manifested through subjective symptoms such as diminished work capacity and impaired attention, which subsequently undermined individuals' emotional regulation abilities. This impairment heightened susceptibility to feelings of helplessness and self-negation, thereby exacerbating depressive tendencies. Basic research has found that fatigue can lead to hypothalamic-pituitary-adrenal axis disorder and neuronal cell death in the hippocampus of rats, accompanied by an increase in depressive behaviors. Numerous applied experimental studies have demonstrated that a decline in positive emotional responses from subjective exercise experience exacerbates daily fatigue [53]. Prolonged exposure to this condition elevates the risk of severe depression and significantly compromises an individual's quality of life [54]. Moreover, Excessive exercise-induced fatigue, particularly when unalleviated by restorative sleep, may amplify emotional distress and perpetuate fatigue accumulation that can reduce exercise motivation and ultimately contribute to the onset of negative affective states, including depression [55]. Therefore, the design of exercise programs should balance moderate physical load with positive psychological experiences to optimize both restorative benefits and subjective enjoyment.

Furthermore, a chain reaction exists between sleep quality and fatigue, jointly mediating the relationship between adolescents' subjective exercise experience and depression, this finding supports Hypothesis 4. Positive exercise experiences contribute to improved sleep quality, diminished fatigue perception, and subsequent reduction in depressive +ymptoms [56]. Fatigue, recognized as a core symptom of insomnia disorders, is associated with mechanisms including excessive sympathetic nervous system activation and frequent nighttime physiological arousal [57]. These factors accelerate metabolic rates, leading to excessive energy depletion and subsequent fatigue symptoms [58]. Quality sleep enhances individuals' energy levels, thereby facilitating stress relief, emotional regulation, cognitive enhancement, and reduction of fatigue symptoms [59,60]. This study empirically examines the mechanism through which subjective exercise

experience influences depressive mood, establishing a chain mediation model that incorporates both sleep quality and fatigue as mediating variables. This study identified sleep quality and fatigue as significant mediators in the relationship between subjective exercise experience and depression. The observed mediation effects elucidate the indirect pathway through which physical activity influences mental health, highlighting the critical role of sleep and fatigue management in emotion regulation. Furthermore, our findings establish a complete chain mediation pathway: subjective exercise experience-sleep quality-fatigue-depression, demonstrating the sequential psychological and physiological mechanisms underlying this association. Based on subjective exercise experience as a predictor variable and sleep quality and fatigue as parallel and continuous mediating variables, the variable path structure was innovated, further revealing the dynamic connection mechanism among behavior (exercise)-physiology (sleep)-subjective energy (fatigue)-emotion (depression), filling the gap in the research of the psychological mechanism of sports behavior. By integrating physiological and psychological mechanisms, this study establishes a comprehensive behavioral-physiological-cognitive-emotional pathway, thereby extending the theoretical framework of emotion regulation. From a subjective experience perspective, our findings reveal that subjective exercise experiences provide superior explanatory power for mental health variations compared to conventional objective metrics (e.g., exercise frequency). This evidence-based approach aligns with and informs practical intervention strategies for adolescent populations.

## 5 Conclusion

The present study demonstrates that positive subjective exercise experience among high school students is significantly correlated with better sleep quality, lower levels of fatigue, and reduced depressive symptoms. Path analysis results provided empirical support for the hypothesis that subjective exercise experience is associated with adolescent depressive symptoms through a sequential mediating pathway involving sleep quality and fatigue. The findings suggest that exercise, when employed as an intervention modality, can systematically enhance positive well-being while mitigating psychological distress, consequently improving sleep quality and alleviating fatigue-all of which collectively reduce depressive tendencies in adolescents. From an implementation standpoint, this research highlights the importance of promoting physical activities that foster positive emotional experiences in adolescent populations, as opposed to focusing solely on quantitative exercise metrics like frequency or intensity. Integrating structured and emotionally engaging exercise interventions into school-based mental health education can significantly bolster adolescents' psychological resilience, mitigate the risk of psychological distress. This approach not only enriches the theoretical foundation for adolescent mental health support systems but also offers a viable and practical pathway for their implementation.

## Supporting information

**S1 File. S1 Dataset.**
(XLSX)

## Author contributions

**Data curation:** Xiaoyu Li.

**Formal analysis:** Xiaoyu Li, Qian Yang, Ziyang Qi.

**Funding acquisition:** Shenghua Qi.

**Investigation:** Qian Yang, Ziyang Qi, Changhao Ma.

**Methodology:** Shenghua Qi.

**Project administration:** Xiaoyu Li.

**Supervision:** Wenliang Ji.

**Validation:** Qian Yang, Changhao Ma, Shenghua Qi.

**Writing – original draft:** Xiaoyu Li.

**Writing – review & editing:** Wenliang Ji.

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
