## [Decision Letter · Decision Letter 0]

21 Jan 2026

Dear Dr. Ji,

Thank you for submitting your manuscript to PLOS ONE. After careful consideration, we feel that it has merit but does not fully meet PLOS ONE’s publication criteria as it currently stands. Therefore, we invite you to submit a revised version of the manuscript that addresses the points raised during the review process.

We look forward to receiving your revised manuscript.

Kind regards,

Tanja Grubić Kezele, Ph.D., M.D.

Academic Editor

PLOS One

Journal Requirements:

“National Social Science Fund of China  grant number [21BTY004]”

6. Please amend either the title on the online submission form (via Edit Submission) or the title in the manuscript so that they are identical.

Additional Editor Comments:

Based on the reviewers' suggestions, the paper needs minor revision. The reviewers' comments can be found below.

Reviewer's Responses to Questions

**Comments to the Author**

1. Is the manuscript technically sound, and do the data support the conclusions?

Reviewer #1: Yes

2. Has the statistical analysis been performed appropriately and rigorously?

Reviewer #1: Yes

3. Have the authors made all data underlying the findings in their manuscript fully available?

Reviewer #1: Yes

4. Is the manuscript presented in an intelligible fashion and written in standard English?

Reviewer #1: Yes

Reviewer #1: The primary theoretical contribution of this study lies in proposing and validating a dual-mediation pathway model demonstrating how subjective movement experiences influence depressive symptoms in adolescents. The findings hold significant practical implications for the prevention and intervention of adolescent depression.

However, the study has significant limitations: its cross-sectional design prevents establishing causal relationships or temporal sequences between variables. While the research identified an association between subjective exercise experiences and depressive symptoms, it cannot determine whether subjective exercise experiences influence depressive symptoms, depressive symptoms affect subjective exercise experiences, or if a bidirectional relationship exists. Second, the sample comprised 538 Chinese high school students, yet the study did not specify the sampling method or sampling frame, making it impossible to assess the representativeness of the sample. Additionally, cultural backgrounds, educational systems, and lifestyles among Chinese adolescents may differ from those in other countries, potentially limiting the generalizability of the findings. Third, the use of self-administered questionnaires for assessment may introduce social desirability bias and recall bias. Particularly when assessing subjective exercise experiences, individual cognitive biases may compromise reporting accuracy. Finally, while the study proposes a dual-mediation pathway model, other crucial mediating variables—such as self-efficacy, social support, and coping strategies—may have been omitted. Furthermore, potential interactions and moderating effects among variables remain under-explored in this complex relationship.

Based on this, it is recommended that corresponding modifications be made!

**Do you want your identity to be public for this peer review?** For information about this choice, including consent withdrawal, please see our For information about this choice, including consent withdrawal, please see our Privacy Policy .

Reviewer #1: **Yes:** Hong Wang.Hong Wang.

---

## [Author Response · Author response to Decision Letter 1]

2 Feb 2026

1.The study has significant limitations: its cross-sectional design prevents establishing causal relationships or temporal sequences between variables. While the research identified an association between subjective exercise experiences and depressive symptoms, it cannot determine whether subjective exercise experiences influence depressive symptoms, depressive symptoms affect subjective exercise experiences, or if a bidirectional relationship exists.

Response: Thank you for your important suggestions. We fully agree with your opinion and have supplemented and revised the content of the manuscript according to your suggestions.

Firstly, since the present study adopted a cross-sectional design, it can only identify the associations among subjective exercise experience, sleep quality, fatigue, and depressive symptoms based on the observed data. Although the proposed chain mediation model revealed significant relationships among these variables, causal inference and temporal ordering cannot be definitively established.

Secondly, through correlation analysis and structural equation modeling, significant direct and indirect paths were observed between subjective exercise experience and depressive symptoms via sleep quality and fatigue. The results showed that adolescents with more positive subjective exercise experiences tended to report better sleep quality, lower levels of fatigue, and fewer depressive symptoms. These findings indicate a consistent and theoretically meaningful pattern of associations, suggesting that subjective exercise experience is closely related to adolescents’ emotional health.

Finally, the identified chain mediation pathway—subjective exercise experience → sleep quality→fatigue→depressive symptoms—aligns well with existing theoretical frameworks and empirical evidence in the literature. From the perspective of the research theme, these findings suggest that interventions emphasizing positive exercise experiences, particularly those that contribute to improvements in sleep quality and reductions in fatigue, may have the potential to alleviate depressive symptoms among adolescents. Nevertheless, we acknowledge that longitudinal or experimental studies are required in future research to further verify the causal direction and stability of these relationships.

We have added a dedicated section on Limitations and Future Directions at the end of the conclusion. In this section, we explicitly note that the cross-sectional design prevents us from establishing causal relationships and acknowledge the potential reverse effects of depressive symptoms on subjective exercise experience and sleep quality. We also suggest that future studies employ longitudinal designs or cross-lagged panel models to address this issue and validate the causal direction of these relationships.

2. The sample comprised 538 Chinese high school students, yet the study did not specify the sampling method or sampling frame, making it impossible to assess the representativeness of the sample. Additionally, cultural backgrounds, educational systems, and lifestyles among Chinese adolescents may differ from those in other countries, potentially limiting the generalizability of the findings.

Response: Thank you for your suggestion. Based on your suggestion, we have made the following explanations.

Firstly, the original submission lacked sufficient details regarding the sampling procedure. We adopted a stratified cluster random sampling method. In the section "2.1 Participants and procedure", we have supplemented a detailed description of the sampling framework.

Thank you for this thoughtful comment. We fully agree that differences in cultural background, educational systems, and lifestyles among Chinese adolescents may limit the generalizability of the findings. We acknowledge this as an important consideration when interpreting the results, and we will explicitly address this issue in future studies by extending the proposed model to samples from different cultural and educational contexts.

3. The use of self-administered questionnaires for assessment may introduce social desirability bias and recall bias. Particularly when assessing subjective exercise experiences, individual cognitive biases may compromise reporting accuracy.

Response: Thank you for this valuable comment. We acknowledge that the use of self-administered questionnaires may introduce potential sources of bias, such as social desirability bias and recall bias, particularly when assessing subjective exercise experiences. Nevertheless, we would like to note that self-report questionnaires remain the most widely used and methodologically feasible approach for assessing psychological constructs such as subjective exercise experience, fatigue, sleep quality, and depressive symptoms. The instruments adopted in the present study have been extensively validated and applied in prior research, demonstrating acceptable reliability and construct validity across adolescent populations. Moreover, subjective exercise experience is inherently perceptual and experiential in nature, making self-report measures an appropriate and commonly accepted method for capturing individual evaluations.

At the same time, we recognize the limitations associated with this assessment approach. Accordingly, we have acknowledged this issue as a methodological limitation and suggest that future studies may benefit from incorporating multi-method assessment strategies, such as objective physical activity indicators, sleep monitoring, or informant-based reports, to further reduce potential reporting bias and enhance measurement accuracy.

4.While the study proposes a dual-mediation pathway model, other crucial mediating variables—such as self-efficacy, social support, and coping strategies—may have been omitted. Furthermore, potential interactions and moderating effects among variables remain under-explored in this complex relationship.

Response: We would like to express our sincere gratitude for your valuable comments and suggestions on our manuscript.

Firstly, this is a very insightful and valuable suggestion. We fully agree that the mechanisms underlying adolescent depression are highly complex, and that a single mediation model is unlikely to capture all relevant psychological, social, and contextual factors involved in this process.

Secondly, in the present study, we intentionally focused on a theoretically grounded and parsimonious chain mediation model involving subjective exercise experience, sleep quality, and fatigue, with the aim of clearly testing a core behavioral–physiological–emotional pathway. To ensure model stability, interpretability, and statistical robustness—particularly given the cross-sectional design—other potentially important variables such as self-efficacy, social support, and coping strategies were not included in the current analysis.

Thirdly, following the reviewer’s suggestion, we have explicitly acknowledged these potentially omitted variables in the Limitations and Future Directions section of the revised manuscript. We now propose that future studies extend the present model by incorporating self-efficacy, social support, and coping strategies as additional mediators or moderators, and by examining possible interaction effects among individual and environmental factors. Such extensions may help construct a more comprehensive and ecologically valid theoretical framework for understanding how subjective exercise experience influences depressive symptoms in adolescents.

---

## [Editor Report · Decision Letter 1]

3 Mar 2026

Unveiling the Chain Mediation Model of Sleep Quality and Fatigue in Linking Subjective Exercise Experience to Depression in Adolescent

PONE-D-25-44933R1

Dear Dr. Ji,

We’re pleased to inform you that your manuscript has been judged scientifically suitable for publication and will be formally accepted for publication once it meets all outstanding technical requirements.

Kind regards,

Tanja Grubić Kezele, Ph.D., M.D.

Academic Editor

PLOS One
---

## [Editor Report · Acceptance letter]

PONE-D-25-44933R1

PLOS One

Dear Dr. Ji,

I'm pleased to inform you that your manuscript has been deemed suitable for publication in PLOS One. Congratulations! Your manuscript is now being handed over to our production team.

Kind regards,

on behalf of

Prof. dr. Tanja Grubić Kezele

Academic Editor

PLOS One